# Determinants of formula feeding among mothers with infants and young children in six Sub Sahara African countries: Multilevel analysis of data from demographic and health survey

**Mohammed Seid Ali**[1]*, **Alebachew Ferede Zegeye**[2], **Belayneh Shetie Workneh**[3], **Gebreeyesus Abera Zeleke**[4], **Enyew Getaneh Mekonen**[4], **Agazhe Aemro**[2], **Berhan Tekeba**[1], **Tadesse Tarik Tamir**[1], **Mulugeta Wassie**[2], **Bewuketu Terefe**[5]

1 Department of Pediatrics and Child Health Nursing, School of Nursing, College of Medicine and Health Sciences, University of Gondar, Gondar, Ethiopia, 2 Department of Medical Nursing, School of Nursing, College of Medicine and Health Sciences, University of Gondar, Gondar, Ethiopia, 3 Department of Emergency and Critical Care Nursing, School of Nursing, College of Medicine and Health Sciences, University of Gondar, Gondar, Ethiopia, 4 Department of Surgical Nursing, School of Nursing, College of Medicine and Health Sciences, University of Gondar, Gondar, Ethiopia, 5 Department of Community Health Nursing, School of Nursing, College of Medicine and Health Sciences, University of Gondar, Gondar, Ethiopia

* muheseid2592@gmail.com

## Abstract

### Introduction

Formula feeding is providing infants with prepared formula as an alternative to or alongside breastfeeding. While breast milk is widely regarded as the optimal source of nutrition for infants, formula feeding is a common practice. The recommended approach is exclusive breastfeeding for the first six months, followed by the introduction of complementary foods after that period, which is crucial for child growth and development. Formal feeding has a negative impact on an infant's health, causing malnutrition and other illnesses. Therefore, this study was investigated to assess formula feeding and determinant factors among mothers with infants in six sub-Saharan African countries.

### Methods

A total weighted sample of 26,119 mothers with infants and young children less than two years was included in this study. The data were taken from a recent demographic and health survey in six sub-Sahara African countries. A multilevel, multivariable logistic regression model was used to identify the determinant factors associated with formula feeding. In the multivariable analysis, the adjusted odds ratio with a 95% CI was used to declare a statistically significant association with formula feeding among mothers with infants.

**Data Availability Statement:** All relevant data are within the manuscript and its Supporting Information files.

**Funding:** The author(s) received no specific funding for this work.

**Competing interests:** The authors have declared that no competing interests exist.

**Abbreviations:** ANC, Antenatal Care; AOR, Adjusted Odd Ratio; CI, Confidence Interval; DHS, Demographic and Health Survey; DHS, Demographic and Health Survey; ICC, Inter-Cluster Correlation; LLR, Log-Likelihood Ratio; MOR, Median Odds Ratio; PCV, Proportion Chang in Variance; SSA, Sub-Sahara Africa; WHO, World Health Organization.

## Results

In this study, the proportion of mothers with infants who use formula feeding was 17.1%. In multilevel logistic analysis (model III), the significant factors associated with formula feeding were the age of the mothers; 25–34 years (AOR = 1.3; 95% CI (1.2–1.41)), 35–49 years (AOR = 1.4; 95% CI (1.22–1.54)), multiple children (AOR = 1.4; 95% CI (1.23–1.77)), maternal educational status; secondary and higher (AOR = 2.4; 95% CI (2.11–2.66)), mother's employment status; (AOR = 1.24; 95% CI (1.14–1.5));, richer households (AOR = 1.2; 95% CI (1.10–1.36)), place of delivery (AOR = 2.1; 95% CI (1.83–2.44)), household media exposure (AOR = 1.5; 95% CI (1.3–1.68))place of residence (AOR = 1.97; 95% CI (1.79–2.17)), community illiteracy level (AOR = 1.17; 95% CI (1.02–1.34)), and community media exposure (AOR = 1.2; 95% CI (1.03–1.38)).

## Conclusion

Formula feeding among mothers with infants in Sub-Saharan Africa has emerged as a public health concern. The recommended approach is to promote exclusive breastfeeding for the first six months, followed by the introduction of complementary feeding after that period. Factors associated with formula feeding include older maternal age, secondary and higher education, delivery in health institutions, employment status, higher household income, twin births, urban residence, low community illiteracy rates, and increased community media exposure. Stakeholders and health policymakers should be focused on strategies to improve breast feeding and discourage infant formula feeding.

## Introduction

Formula feeding is feeding an infant or young child prepared formula instead of or in addition to breastfeeding. It is formulated industrially following standards for infants, usually prepared for bottle-feeding from powder or liquid [1]. Formula feeding is indicated when the mother has an illness that could be passed on to the baby through breast milk or through the close physical proximity required for breastfeeding. Otherwise, experts agreed that breastfeeding is the best infant nutrition [2–4]. Several studies have shown that breast milk contains a variety of bioactive nutrients that help the function of the gastrointestinal tract, brain development, and immune system. Thus, breast milk is recognized as a biological fluid required for optimal infant growth and development [5].

The World Health Organization (WHO) and other organizations like UNICEF recommend that infants be exclusively breastfed for their first 6 months of life and then introduced to complementary feedings with breastfeed from 6 months to 23 months [6]. Exclusive breastfeeding for six months is the basic period for child growth, development, and overall health because breast milk provides essential, complete, and comprehensive nutrition for the child [7]. Breast milk protects mothers against certain types of health conditions, like bleeding after delivery, breast and ovarian cancer, and it also delays the next pregnancy. Furthermore, breast milk provides infants with important nutrients and antibodies that lower the risk of malnutrition and diarrheal illnesses, which are among the leading causes of infant mortality in low and middle-income countries [8–10]. Previous studies showed that exclusive breastfeeding is decreasing

and being replaced by formula, including plain water, fruit juice, and other local homemade foods [11, 12].

Globally, around two out of five mothers introduced breast milk substitutes (formula feeding) by the time their baby was eight weeks old, and most were combining breast milk and formula substitutes before their baby reached six months of age. Only 40% of mothers exclusively breastfeed their children for six months globally. Although the rates of exclusive breast feeding have been rising for the past two decades, it is still a long road to achieve the 100% global target coverage recommended by UNICEF. Currently, exclusive breast feeding is low in the developing world, particularly in West and Central Africa, which happen to have one of the highest rates of infant malnutrition in the world [13, 14]. The prevalence of exclusive breastfeeding rates in Sub-Saharan Africa (SSA) is 35%, which is lower in comparison to other low- and middle-income countries (39%) [15]. In developing countries, including SSA countries, women discard colostrum due to traditional or cultural beliefs, perceiving it as sour, difficult to digest, and harmful to the infant's health. Thus, they replace it with pre-lacteal feedings such as plain water, honey, formula, or animal milk [16, 17]. Evidence showed that the exclusive breastfeeding rate increased in all areas except for the Middle East and Africa. Formula consumption rates in the first six months of life increased in SSA and South Asia [18].

Formula feeding has many unwanted side effects that can result in adverse outcomes for the infant's health. The higher protein content of artificial baby formula compared to the lower protein content in breast milk is responsible for the increased growth rate and adiposity during the influential period of the infancy of formula-fed infants, which leads to obesity and related problems [19]. Infants who are exposed to formula feeding have higher risks of illness, allergies, obesity, sudden infant death syndrome, and impairment of the child's cognitive development. There is also an increased risk of long-term diseases with an immunological basis, including asthma and other atopic conditions. Formula feeding is also associated with a greater risk of childhood leukemia [20, 21]. In 2012, the World Health Assembly endorsed a comprehensive implementation plan on maternal, infant, and young child nutrition, specifying six global nutrition targets for 2025, one of which is to increase the rate of exclusive breastfeeding in the first six months up to at least 50% [22]. In SSA, where child morbidity and mortality are very high due to diarrhea and other infections, exclusive breastfeeding is the best child-feeding option compared to formula or other feeding practices [23].

Despite the fact that formula feeding practice is increasing in SSA countries, the exact prevalence and determinant factors of formula feeding are not known. There are very limited studies about formula feeding that are not enough to provide comprehensive evidence for policymakers and stakeholders. Thus, this study will help to provide customized evidence-based support for mothers to prevent social, personal, and commercial influences that lead to their decision to feed infant formulas, and it will also help the responsible bodies to design strategies promoting exclusive breast feeding.

## Methods

### Study area and period

The data were pooled from six SSA countries, namely Ethiopia, Somalia, Cameroon, Gambia, Ghana, and South Africa, from 2018–2022. DHSs are nationally representative household surveys that provide data from a wide range of indicators in the areas of health, population, and nutrition with face-to-face interviews of reproductive-aged women.

## Study variables

The dependent variable in this study was formula feeding, which was derived from the DHS question, "Have you ever given a formula feeding to your baby?" The outcome variable was dichotomized as "yes" if a woman had given formula feeding for her baby and "no" if a woman didn't give formula feeding within the study period. The independent variables included in the study were maternal age, marital status, maternal education, maternal employment status, wealth status, family size, age of child, sex of child, plurality of birth, breast feeding status, place of delivery, ANC visits, preceding birth interval were individual level variables, and place of residence, community level women education, community mass media exposure, and community level poverty were community level variables.

## Data source, extraction, and study participants

In this study, the data was obtained from the MEASURE Demographic and Health Survey (DHS) program. The survey aimed to provide up-to-date information on socio-economic, demographic, nutrition, and health indicators to plan, monitor, and evaluate various health programs and policies. The clusters were developed through a process of household listing and georeferencing. Interviews were conducted only in the pre-selected households and clusters; no replacement of the pre-selected units was allowed during the survey data collection stages. For this study, we used the Kids Record (KR) data set with a total weighted sample of 26,119 women who ever breastfed and who had children less than two years of age.

## Data management and multilevel analysis

Appending data, extraction of data, re-coding, and statistical analysis were carried out using STATA version-17 software. The sample weights were applied to compensate for the unequal probability of selection between the strata and to keep the sample representative. Because of the hierarchical and clustered nature of the DHS data, multilevel analysis was conducted. A multilevel, multivariable logistic analysis was employed to identify the determinant factors associated with formula feeding. Both bivariable and multivariable logistic analyses were conducted. Variables with a p-value < 0.2 in the bivariable analysis were eligible for multivariable analysis. In the multivariable analysis, adjusted odds ratio (AOR) with a 95% confidence interval (CI) was reported, and variables with a p-value < 0.05 were declared to be statistically significant factors for formula feeding.

## Model building

While conducting multilevel analysis, four models were fitted. The null model was a model without explanatory variables, which was used to determine random effects at the overall population level and assess the heterogeneity in the community. Model I was a model with individual-level explanatory variables only; model II was a model with community-level variables only; and model III was a model with both individual- and community-level variables.

## Parameter estimation

For examining the cluster-level variability of formula feeding, we have employed random effect analysis. Calculating the intra-class correlation coefficient (ICC), deviance, proportional change in variance (PCV), median odds ratio (MOR) and log likelihood ratio (LLR) were used as indicators of heterogeneity. The degree of heterogeneity of formula feeding between clusters was quantified by ICC, calculated as ICC = VC/(VC = 3.29)*100%. The MOR quantifies the variation in formula feeding between clusters in terms of the odds ratio scale and is calculated as

MOR = e0.95 $\sqrt{}$variance. Moreover, PCV demonstrates the variation in formula feeding explained by the determinants computed as PCV = (Vnull-Vc)/Vnull*100%; where Vnull is the variance of the null model and Vc is the cluster-level variance. Due to the nested nature of the models, deviance and LLR were used for model comparison. The model with the lowest deviance and highest LLR was considered the best fit. To assess multicollinearity, the variance inflation factor (VIF) was used, and there was no multicollinearity between independent variables, with a mean VIF of 1.51 (the minimum and maximum VIF were 1.02 and 3.28, respectively).

## Ethical approval and consent to participate

This study was based on secondary data analysis of publicly available national survey data from the DHS program. Ethics approval permission was obtained from MEASURE DHS program to use the data set for this study. We requested permission to download the DHS program, and it was granted. It uses data from http://www.dhsprogram.com. Before data collection, the DHS program approved that written or verbal consent were obtained from participants. The Institutional Review Board and the Research and Ethics Committee of the Kenyan Ministry of Health gave their approval to the survey protocol. All procedures were carried out in accordance with the Declaration of Helsinki on ethical principles for conducting human research, in addition to receiving ethics approval and informed consent.

## Results

### Socio-demographic-related characteristics of the participants

A total of 26,119 mothers with infants and young children participated in this study from six SSA countries. Among the total, nearly one-third (33.9%) of the children were in the category of 24–59 months, and more than half (53.57%) of the women were in the age group of 25–34 years. Regarding maternal educational status, about 44.87% of the total participants had attended secondary and higher levels of education. About 63.13% of the mothers were married. Among the total participants, more than half (62.03%) of the mothers were unemployed. These demographic details provide valuable context for understanding the study's results and implications (Table 1).

### Socio-economic and health care-related characteristics of the participants

Among the total participants, about 31.29% of the respondents were from the poorest households in terms of their wealth index status. Regarding the place of delivery, the majority (74.6%) of the mothers delivered in health institutions. About one-fourth (24.19%) of the participant's preceding birth interval was less than twenty-four months. About 42.07% and 44.55% of the participants had media exposure through radio and television, respectively. Regarding the ANC follow-up, about 46.49% of the women had more than four ANC follow-up visits. The socio-economic and healthcare variables provide further context for understanding maternal and child health practices in the study population (Table 2).

### Prevalence of formula feeding in six SSA countries

In this study, the overall prevalence of formula feeding in six SSA countries was 17.1%, with a 95% CI (16.6%-17.5%) (Fig 1). The prevalence of formula feeding for infants and young children varies from one country to another. Among the six SSA countries included in this study, the prevalence of formula feeding was highest in Gabon (38.5%) and Zambia (35.42%), while Ethiopia (4.39%) and Cameroon (9.42%) had relatively lower prevalence's of formula feeding. These variations highlight the importance of context-specific interventions and policies to

**Table 1. Socio-demographic related characteristics of the participants in six SSA countries (n = 26,119).**

| Variables | Response | Frequency | Percent (%) |
|---|---|---|---|
| Age of mother in years | 15–24 | 9,319 | 35.68 |
| | 25–34 | 13,991 | 53.57 |
| | 35–49 | 2,809 | 10.75 |
| Current age of child | < 6 months | 5,405 | 20.69 |
| | 6–23 months | 11,860 | 45.41 |
| | 24–59 months | 8,854 | 33.90 |
| Plurality of child | Single | 25,194 | 96.46 |
| | Twin | 925 | 3.54 |
| Sex of child | Male | 13,345 | 51.09 |
| | Female | 12,774 | 48.91 |
| Place of residence | Urban | 10,419 | 39.89 |
| | Rural | 15,700 | 60.11 |
| Mother's education | No education | 8,432 | 32.28 |
| | Primary education | 5,967 | 22.85 |
| | Secondary and higher education | 11,720 | 44.87 |
| Marital status | Unmarried | 3458 | 13.24 |
| | Married | 16,490 | 63.13 |
| | Ever married | 6,171 | 23.63 |
| Mother's employment status | Unemployed | 9,918 | 37.97 |
| | Employed | 16,201 | 62.03 |

promote optimal feeding practices in SSA countries. Breastfeeding initiatives should consider cultural factors and health-service access, especially for mothers from low socio-economic backgrounds (Fig 2 and Table 3).

## Random effect analysis for cluster variability and model fitness

The total variation in the prevalence of formula feeding among infants was attributable to clustering. The clustering effect and model estimation are shown in Table 4. The result of the null model showed that there was significant variability in the odds of practicing formula feeding, with a community variance of 0.49. In addition, the MOR was 1.81, meaning that the odds of practicing formula feeding might increase when respondents moved from low-to high-risk communities. This indicated the existence of significant heterogeneity in formula feeding across different clusters. This variability is directed toward conducting multilevel analysis to identify the determinant factors associated with formula feeding among infants in six SSA countries. Model III was the best-fitted model, with the lowest deviance and the highest LLR. In model III (adjusted for both individual and community variables), the variance (0.34) remained significant (p-value <0.001). The PCV in this model was 29.9%, which shows that 29.9% of the cluster variance observed in the null model was explained by both individual-level and community-level variables (Table 4).

## A multilevel analysis of the factors associated with formula feeding in six SSA countries

The multivariable logistic regression was carried out to identify factors associated with formula feeding and was presented with an AOR and 95% CI. In model I, the determinant factors were identified at the individual/household level. The significant factors associated with formula feeding at the individual level were the age of the women; 25–34 years (AOR = 1.25; 95% CI

**Table 2. Socio-economic and health care related variables of the participants in six SSA countries (n = 26,119).**

| Variables | Response | Frequency | Percent (%) |
|---|---|---|---|
| Wealth index | Poorest | 8,172 | 31.29 |
| | Poorer | 5947 | 22.77 |
| | Middle | 4,973 | 19.04 |
| | Richer | 3,903 | 14.94 |
| | Richest | 3,124 | 11.96 |
| Place of delivery | Home | 6,613 | 25.32 |
| | Health institutions | 19,506 | 74.68 |
| Preceding birth interval | <24 months | 6,319 | 24.19 |
| | ≥24 months | 19,800 | 75.81 |
| Birth order | One birth | 6634 | 25.40 |
| | Two-three births | 10,025 | 38.38 |
| | Four or more births | 9,460 | 36.22 |
| ANC visits | No visit | 2,438 | 9.33 |
| | <4 visits | 11,538 | 44.17 |
| | ≥4 visits | 12,143 | 46.49 |
| The size of child at birth | Small | 7,305 | 27.97 |
| | Average | 10,406 | 39.84 |
| | Large | 3,617 | 13.85 |
| Radio exposure | No | 15,130 | 57.93 |
| | Yes | 10,989 | 42.07 |
| Television exposure | No | 14,484 | 55.45 |
| | Yes | 11,635 | 44.55 |
| Reading newspaper/magazine | No | 22,029 | 84.34 |
| | Yes | 4,090 | 15.66 |

(1.15–1.37)), 35–49 years (AOR = 1.33; 95% CI (1.18–1.49)), plurality of children (AOR = 1.54; 95% CI (1.28–1.84)), maternal education; secondary and higher (AOR = 2.2; 95% CI (1.94–2.45)), mother's employment status (AOR = 1.27; 95% CI (1.17–1.37)), wealth index; richer place of delivery (AOR = 2.25; 95% CI (1.95–2.61)), and household media exposure (AOR = 1.63; 95% CI (1.46–1.82)). At the community level, the significant factors associated with formula feeding were place of residence; urban residents (AOR = 3.7; 95% CI (3.4–4.06)), community illiteracy level (AOR = 1.4; 95% CI (1.2–1.6)), and community media exposure (AOR = 1.3; 95% CI (1.2–1.5)).

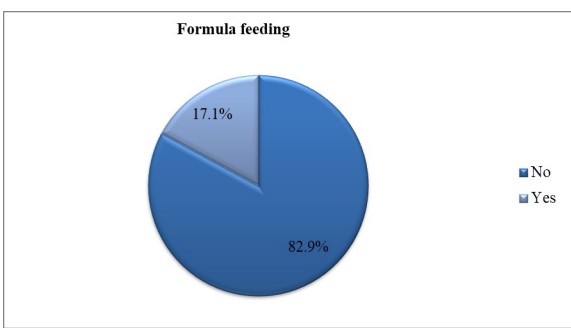

**Fig 1. The overall prevalence of formula feeding in six SSA countries (n = 26,119).**

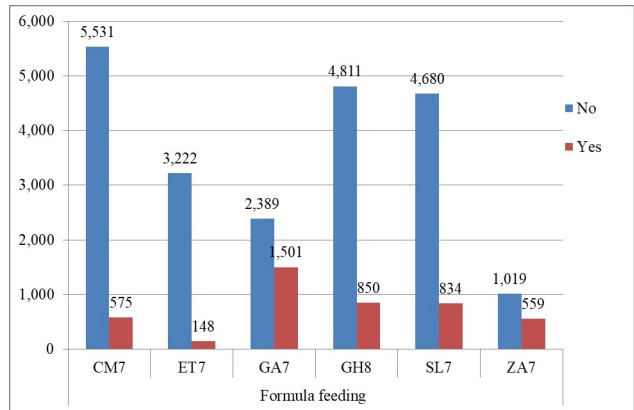

**Fig 2. The country distribution of formula feeding prevalence among six Sub Sahara Africa countries (n = 26,119).**

The model with the smaller deviance and the largest LLR (model III) was the best fit, and the interpretation of the results was based on this model. In multivariable logistic analysis (model III), the significant factors associated with formula feeding were the age of the mothers: 25–34 years (AOR = 1.3; 95% CI (1.2–1.41)), 35–49 years (AOR = 1.4; 95% CI (1.22–1.54)), plurality of child (AOR = 1.4; 95% CI (1.23–1.77)), maternal educational status; secondary and higher (AOR = 2.4; 95% CI (2.11–2.66)), mother's employment status; (AOR = 1.24; 95% CI (1.14–1.5)), richer households (AOR = 1.2; 95% CI (1.10–1.36)), place of delivery (AOR = 2.1; 95% CI (1.83–2.44)), household media exposure (AOR = 1.5; 95% CI (1.3–1.68)), place of residence (AOR = 1.97; 95% CI (1.79–2.17)), community illiteracy level (AOR = 1.17; 95% CI (1.02–1.34)), and community media exposure (AOR = 1.2; 95% CI (1.03–1.38)) (Table 5).

## Discussion

This study investigated the prevalence of formula feeding and its determinant factors among mothers with infants and young children in six SSA countries using recent data from DHS 2018–2022. The prevalence of formula feeding among mothers with infants and young children in six SSA countries was 17.10%, with a 95% CI of 16.6%-17.5%. Based on this finding, the prevalence of formula feeding utilization for infants increased and became a common practice, while exclusive breast feeding is the single most important way of feeding infants. Formula feeding practice was affected by socio-demographic, socio-economic, and health care service-related factors. Studying the status of formula feeding is an important indicator of child health care quality and the healthcare system for future improvements. Formula feeding is an alternative infant feeding mechanism when the mother cannot breastfeed due to illness,

**Table 3. The prevalence of formula feeding distribution by country in six SSA countries (n = 26,119).**

| Country | Survey year | Infant formula feeding status | | Percentage of formula feeding |
|---|---|---|---|---|
| | | **No** | **Yes** | |
| Cameroon | 2018 | 5,531 | 575 | 9.42% |
| Ethiopia | 2019 | 3,222 | 148 | 4.39% |
| Gabon | 2019/20 | 2,389 | 1,501 | 38.59% |
| Ghana | 2022 | 4,811 | 850 | 15.02% |
| Somalia | 2019 | 4,680 | 834 | 15.13% |
| South Africa | 2016 | 1,019 | 559 | 35.42% |

**Table 4. Random effect analysis for cluster variability and model fitness for assessing formula feeding among infants in six SSA countries (n = 26,119).**

| Parameter | Null model | Model I | Model II | Model III |
|---|---|---|---|---|
| Variance | 0.4912641 | 0.3467725 | 0.3610106 | 0.3439758 |
| ICC | 0.1299251 | 0.0953552 | 0.0988832 | 0.0946589 |
| MOR | 1.81 | 1.52 | 1.55 | 1.51 |
| PCV | Reference | 29.41% | 26.5% | 29.9% |
| **Model fitness** | | | | |
| LLR | -11607.692 | -9164.5069 | -10958.632 | -9140.8322 |
| Deviance | 23,215.384 | 18,329.0138 | 21,917.264 | 18,281.6644 |

**Table 5. Multilevel analysis for the determinant factors associated with formula feeding in six SSA countries (n = 26,119).**

| Variables | Response | Model I AOR (95% CI) | Model II AOR (95% CI) | Model III AOR (95% CI) |
|---|---|---|---|---|
| Age of mother in years | 15–24 | 1.0 | - | 1.0 |
| | 25–34 | 1.25(1.15–1.37) | - | 1.3(1.2–1.41)* |
| | 35–49 | 1.33(1.18–1.49) | - | 1.4(1.22–1.54)* |
| Plurality of child | Single | 1.0 | - | 1.0 |
| | Twin | 1.54(1.28–1.84) | - | 1.4(1.23–1.77)* |
| Sex of child | Male | 1.0 | - | 1.0 |
| | Female | 1.04(0.96–1.12) | - | 1.03(0.93–1.11) |
| Mother's education | No education | 1.0 | - | 1.0 |
| | Primary | 0.9(0.8–1.01) | - | 0.9(0.8–1.08) |
| | Secondary/higher | 2.2(1.94–2.45) | - | 2.4(2.11–2.66)* |
| Mother's employment status | Unemployed | 1.0 | - | 1.0 |
| | Employed | 1.27(1.17–1.37) | - | 1.24(1.14–1.5)* |
| Wealth status | Poorer | 1.0 | - | 1.0 |
| | Middle | 0.9(0.87–1.08) | - | 0.9(0.81–1.01) |
| | Richer | 1.5(1.35–1.64) | - | 1.2(1.10–1.36)* |
| Place of delivery | Home | 1.0 | - | 1.0 |
| | Institutions | 2.25(1.95–2.61) | - | 2.1(1.83–2.44)* |
| The size of child at birth | Small | 1.0(0.92–1.15) | - | 1.0(0.93–1.16) |
| | Average | 0.99(0.91–1.01 | - | 1.0(0.93–1.10) |
| | Large | 1.0 | - | 1.0 |
| Current age of child | Less 6 months | 1.17(0.91–1.52 | - | 1.1(0.98–1.22) |
| | 6–23 months | 1.1(0.93–1.23 | - | 1.13(0.93–1.44) |
| | 24–59 months | 1.0 | - | 1.0 |
| Household media exposure | No | 1.0 | - | 1.0 |
| | Yes | 1.63(1.46–1.82) | - | 1.5(1.3–1.68)1* |
| Place of residence | Urban | - | 3.7(3.4–4.1) | 1.97(1.79–2.17)* |
| | Rural | - | 1.0 | 1.0 |
| Community illiteracy level | High | - | 1.0 | 1.0 |
| | Low | - | 1.4(1.2–1.6) | 1.17(1.02–1.34)* |
| Community poverty level | High | - | 1.2(1.0–1.35) | 1.11(0.97–1.28) |
| | Low | - | 1.0 | 1.0 |
| Community mass media exposure | High | - | 1.3(1.2–1.5) | 1.2(1.03–1.38)* |
| | Low | - | 1.0 | 1.0 |

*statistically significant at a p-value <0.05

death, or any other unsuitable conditions for breastfeeding. Unfortunately, it has become a commonly performed feeding practice without reason in developing countries like the SSA.

The prevalence of formula feeding in this study was lower than in other studies conducted in Egypt [24] and Saudi Arabia [25]. A systematic review and meta-analysis study showed that the prevalence of mixed feeding before six months was 36% in the Middle East and Africa together [26]. This might be due to the fact that the data in this study was national DHS data; the survey sample includes all populations in the country. The prevalence of formula feeding might be lower in large-sample studies than in basic studies [27, 28]. Formula feeding practice in this study was higher than in other studies conducted in London, England, 13% [29]. The possible explanation for this discrepancy might be due to the variation in the health care system and policy between African countries and England. At the individual country level, the prevalence of formula feeding was highest in Gabon (38.59%), South Africa (35.42%), and Ethiopia (4.39%). This indicates that there is variation in formula feeding practices from one country to another because formula feeding is affected by socio-demographic, socio-economic, and health care service-related variables.

This study identified the determinants associated with formula feeding. The significant factors associated with formula feeding utilization were infants with older mothers, mothers having secondary and higher education, delivery in the health institutions, employed mothers, mothers from richer households, mothers who had twin infants, urban residents, mothers who live in low levels of community illiteracy, and high levels of community media exposure.

In this study, maternal age was significantly associated with infant formula feeding practices. Those mothers older than 25 years were more likely to utilize formula feeding as compared to those younger. This finding was evidenced by another study [30]. This could also be due to the fact that maternal-related complications might be higher as the age of women increases. Many maternal-related health problems arise among women of older age, which might be the reason not to breastfeed exclusively and lead them to utilize formula feeding for their infants. Medical conditions like HIV, hepatitis, and any diseases that can affect safe breast feeding practices might necessitate formula feeding practices. These medical conditions could be higher among older mothers as compared to younger mothers [31]. Each woman's situation is unique, and decisions about breastfeeding or formula feeding depend on various factors. Health care professionals assess the risks and benefits, considering both maternal health and infant well-being [32].

The result of this study showed that the plurality of children (twin childbirth) affects the utilization of infant formula feeding. Those mothers who had twin babies had higher odds of utilizing infant formula feeding as compared to single babies. This finding was consistent with other studies [33, 34]. This might be due to the fact that twin infants require more breast milk to provide food. This situation makes it difficult to provide exclusive breast feeding and might not be enough for twin or triple babies, which leads mothers to utilize formula feeding. In addition, twin babies require more calories, protein, vitamins, and minerals for healthy growth and development. Breast milk production needs to match these increased demands, which can be challenging for mothers [35].

The employment status of mothers was associated with infant formula feeding. Those mothers with infants who were employed had higher odds of utilizing infant formula feeding as compared to unemployed mothers; this finding was supported by other studies [36, 37]. This might be due to the fact that employed mothers could face challenges in breastfeeding their infants because they are at work. It might also be that employed mothers did not get adequate time to breastfeed their infants and found it difficult to breastfeed in the workplace instead; they utilized formula feeding for their infants. Balancing work responsibilities and

breastfeeding can indeed be challenging for employed mothers. Employers must allow a reasonable break time for breastfeeding for up to one year after the child's birth [38].

Maternal educational status was significantly associated with formula feeding. The odds of formula feeding practice were higher among educated mothers as compared to mothers who had no formal education. This finding was supported by other studies [39, 40]. Formula feeding was higher among participants with a low level of community illiteracy. This means that a higher level of maternal education in the community was linked to higher odds of formula feeding utilization for their infants as compared to an uneducated community. Educated mothers often have better access to information about infant feeding options. They might be more aware of the benefits and potential drawbacks of both breastfeeding and formula feeding. Many educated mothers have demanding jobs that make breastfeeding challenging. Formula feeding can be more convenient, especially if they need to return to work soon after childbirth. Similar studies proved this finding [41, 42]. This might also be because educated mothers assume that formula feeding helps the infant grow early and is important for child care. But the recommended infant feeding practice is exclusive breast feeding for the first six months. The community literacy level may affect individual feeding practices. The possible explanation for this situation might be that community literacy influences parents' knowledge about nutrition, feeding plans, and feeding practices. In communities with higher literacy levels, parents may make more informed decisions regarding their children's feeding [43].

The wealth status of the households was another important factor associated with formula feeding. Mothers from richer households had higher odds of feeding infant formula to their children as compared to their counterparts. This finding was consistent with other studies [44]. This might be due to mothers with higher wealth status being able to access basic health care services in private settings, covering the cost of transportation, buying commercially prepared infant formulas, and accessing infant formula even if it is expensive.

This study showed that place of residence affects infant formula feeding practice. The odds of formula feeding were higher among infants and young children from urban areas as compared to rural areas. This finding was supported by other studies [42, 45]. This might be due to the fact that infant formula milk is mainly available commercially in urban areas rather than rural areas. Cultural beliefs affect infant feeding practices [46]. Some mothers in urban areas usually utilize formula feeding for aesthetic purposes, there's a belief that breastfeeding mothers may appear older and experience breast sagging. However, it's essential to recognize that these perceptions are subjective and not necessarily based on scientific evidence [47].

The place of delivery was significantly associated with infant formula feeding. Those mothers who deliver in health institutions were 2.1 times more likely to utilize infant formula feeding as compared to those with home delivery. Delivery in the health institution is usually linked to high utilization of formula feeding. Some hospitals may provide formula to newborns, especially if there are concerns about the mother's milk supply or if the baby has difficulty breastfeeding initially. Formula feeding can be seen as a more convenient option for new mothers, especially if they are recovering from childbirth and need rest [48, 49]. The mode of delivery in the health institutions might also be a cesarean section; in this case, formula feeding might be practiced for the infants. While breastfeeding is generally recommended due to its numerous benefits, there are situations where formula feeding becomes necessary. This finding was in agreement with other studies [50, 51]. This might be due to the fact that high-risk pregnancies are usually admitted to health institutions for delivery, and maternal complications might require to utilize infant formula feeding [52].

Household media exposure was one of the determinant factors associated with formula feeding. Those mothers with infants who lived in highly media-exposed households were 1.5 times more likely to practice infant formula feeding as compared to their counterparts. This

finding was supported by other studies [53–55]. The mass media is important to distribute health-related information to the large population in the country. It helps to promote appropriate infant and young child feeding, improve the habit of exclusive breast feeding, and discourage formula feeding. Mass media plays an important role in providing information the country. This study also proved that mass media exposure at the community level affects formula feeding practices. Mothers in a community of highly mass media exposure had higher odds of utilizing formula feeding as compared to their counterparts. On the contrary, few studies showed that high media exposure promotes appropriate child feeding and exclusive breast feeding. While media exposure can shape perceptions and behaviors, there isn't direct evidence that high media exposure specifically promotes exclusive breastfeeding. Media exposure, particularly through advertising, has a significant impact on infant feeding practices. The manufacturers of formula milk promote their product through media, formula feeding, often presenting it as a convenient and modern alternative to breastfeeding. Studies have shown that mass media content, especially advertisements for infant formula, can discourage breastfeeding by promoting formula feeding as a modern and socially desirable choice for commercial purpose [56, 57].

However, media campaigns and educational content can raise awareness about its benefits [58]. It needs further study including, qualitative research methods, to understand how the mass media affects child feeding practices. Future researchers will address this issue. Generally, infant formula feeding should not be used unless there is difficult situation to promote exclusive breast feeding for the first six months [59].

## Limitation of the study

Since this study was conducted from DHS data, some variables like clinical, maternal, and obstetric-related factors that might affect formula feeding were not included, and future researchers can conduct studies including these variables.

## Conclusion

In sub-Saharan Africa, formula feeding has become more common among mothers with infants and young children. However, it's important to note that the recommended feeding practice is exclusive breastfeeding for the first six months, followed by complementary feeding after that period. The likelihood of formula feeding utilization was higher among older mothers, mothers having secondary and higher education, delivery in the health institutions, employed mothers, mothers from richer households, mothers who had twin infants, urban residents, mothers who live in low levels of community illiteracy, and high levels of community media exposure. Therefore, policymakers and other stakeholders, such as governmental and non-governmental organizations, should focus on strategies to improve appropriate infant feeding practices and promote breastfeeding rather than formula feeding. The above determinant factors are an important input to developing strategies to promote appropriate infant feeding practices in the SSA countries.

## Supporting information

**S1 Data.**
(XLS)

## Acknowledgments

The authors acknowledged the DHS program managers.

## Author Contributions

**Conceptualization:** Mohammed Seid Ali, Enyew Getaneh Mekonen, Berhan Tekeba.

**Data curation:** Mohammed Seid Ali, Belayneh Shetie Workneh, Gebreeyesus Abera Zeleke, Agazhe Aemro, Tadesse Tarik Tamir, Mulugeta Wassie.

**Formal analysis:** Mohammed Seid Ali, Belayneh Shetie Workneh, Enyew Getaneh Mekonen.

**Funding acquisition:** Mohammed Seid Ali, Gebreeyesus Abera Zeleke, Enyew Getaneh Mekonen, Agazhe Aemro, Berhan Tekeba, Bewuketu Terefe.

**Investigation:** Mohammed Seid Ali, Berhan Tekeba.

**Methodology:** Mohammed Seid Ali, Belayneh Shetie Workneh, Enyew Getaneh Mekonen.

**Project administration:** Mohammed Seid Ali, Alebachew Ferede Zegeye, Belayneh Shetie Workneh, Gebreeyesus Abera Zeleke, Enyew Getaneh Mekonen, Agazhe Aemro, Berhan Tekeba, Tadesse Tarik Tamir, Mulugeta Wassie, Bewuketu Terefe.

**Resources:** Alebachew Ferede Zegeye, Tadesse Tarik Tamir.

**Software:** Alebachew Ferede Zegeye, Gebreeyesus Abera Zeleke, Agazhe Aemro, Mulugeta Wassie.

**Supervision:** Alebachew Ferede Zegeye, Mulugeta Wassie.

**Validation:** Mohammed Seid Ali, Alebachew Ferede Zegeye, Enyew Getaneh Mekonen, Agazhe Aemro, Tadesse Tarik Tamir, Bewuketu Terefe.

**Visualization:** Mohammed Seid Ali, Alebachew Ferede Zegeye, Gebreeyesus Abera Zeleke, Enyew Getaneh Mekonen, Bewuketu Terefe.

**Writing – original draft:** Mohammed Seid Ali, Belayneh Shetie Workneh, Tadesse Tarik Tamir, Bewuketu Terefe.

**Writing – review & editing:** Mohammed Seid Ali, Belayneh Shetie Workneh, Berhan Tekeba, Mulugeta Wassie, Bewuketu Terefe.

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
