## [Decision Letter · Decision Letter 0]

29 Jul 2024

PONE-D-24-24055Determinants of formula feeding among mothers with infants and young children in six Sub Sahara African countries: Multilevel analysis of data from demographic and health surveyPLOS ONE

Dear Dr. Ali,

Thank you for submitting your manuscript to PLOS ONE. After careful consideration, we feel that it has merit but does not fully meet PLOS ONE’s publication criteria as it currently stands. Therefore, we invite you to submit a revised version of the manuscript that addresses the points raised during the review process.

**ACADEMIC EDITOR: **What is the rationale behind choosing multilevel analysis? The study spans a wide geographical area across six Sub-Saharan African countries, where data consistency varies. It is important to address contextual effect modifications and confounders that could influence the outcomes. These factors need careful consideration in the analysis.Including a map to illustrate the geographical coverage would be beneficial for the paper, though it is not mandatory.Please revise the keywords according to MeSH (Medical Subject Headings) terms.Please address each reviewer's comments one by one.==============================

We look forward to receiving your revised manuscript.

Kind regards,

Dev Ram Sunuwar, MS

Academic Editor

PLOS ONE

Reviewers' comments:

Reviewer's Responses to Questions

**Comments to the Author**

1. Is the manuscript technically sound, and do the data support the conclusions?

Reviewer #1: Partly

Reviewer #2: No

Reviewer #3: Yes

Reviewer #4: Yes

Reviewer #5: No

2. Has the statistical analysis been performed appropriately and rigorously? 

Reviewer #1: Yes

Reviewer #2: No

Reviewer #3: Yes

Reviewer #4: Yes

Reviewer #5: Yes

3. Have the authors made all data underlying the findings in their manuscript fully available?

Reviewer #1: Yes

Reviewer #2: No

Reviewer #3: Yes

Reviewer #4: Yes

Reviewer #5: Yes

4. Is the manuscript presented in an intelligible fashion and written in standard English?

Reviewer #1: Yes

Reviewer #2: Yes

Reviewer #3: Yes

Reviewer #4: Yes

Reviewer #5: No

5. Review Comments to the Author

Reviewer #1: Thank you for reviewing the DHS findings from six countries. Some of your conclusions appear to be misleading, as statistical significance does not always align with real-world outcomes. For instance, factors such as maternal education, institutional delivery, and exposure to media were identified as significantly linked to bottle feeding. However, is this truly justifiable in practical terms, especially considering our emphasis on promoting and prioritizing institutional deliveries?

In the discussion section, while you have referenced supportive articles to support your findings, it would be prudent to also include articles that present contrary evidence against your statistical conclusions.

Reviewer #2: The authors have analyzed the key factors associated with formula feeding among mothers with infants in six sub-Saharan African (SSA) countries. The data for the analyses come from the MEASURE Demographic and Health Survey program - specifically the Kids Record (KR) subset. The authors have performed multi-level multivariable analysis to identify the determinant factors at both individual and community levels.

As a reviewer, I did not have a hard time reading the manuscript or understanding the sequence of arguments presented in (a) justifying the necessity of such a study, and (b) substantiating the model choices and the results thereafter. However, in a major portion of the manuscript, the arguments themselves are inadequate, and the analyses seem almost mechanical and brute-force, allowing little room for data exploration and judgement that is a fundamental part of any statistical analysis workflow.

As such, the manuscript is problematic in its current form for me in both methodological and scientific ways, and I hope my point-by-point queries presented below will help the authors to formulate the paper in an improved, sound fashion.

1. The authors summarize the study variables in Tables 1, 2, and 3, but spend very little time in trying to assess what those percentages mean. Lines 176-196 very briefly touch upon some of these percentages, but fail to mention some key aspects.

1.a. There are some categorical variables that are extremely imbalanced - such as child plurality (almost a 96 to 4 breakup). Are such variables even fit to be included in any model?

1.b. The authors absolutely ignore the fact that the overall summaries in Tables 1-2 do not represent any potential country-specific pattern. What if a certain country has higher or lower proportions of a particular age group of mothers, or differences across education or marital status? Do we know that all SSA countries are uniform in these terms? Only a country-specific summary of each variable will answer this - and only after that we can even talk about which variables are worth including in any candidate model at all.

2. The modeling scheme seems rushed and in some places, odd. There are quite important questions that the authors do not even attempt to answer.

2.a. Table 3 clearly shows that formula feeding itself is not consistent across the countries. Gabon is a huge outlier. Ghana and Somalia are similar, with Cameroon and Ethiopia showing much smaller percentages. It is extremely possible that many of the associations identified in the models are country-specific variations in demographics and health indicators masked as true determinants. Whether there is country-specific effect modification, or confounding of any kind is a question that needs to be answered before making any kind of conclusive statement.

2.b. The authors mention that variables with a p-value < 0.2 in a bivariate analysis were included in the multivariate models. In the three non-null models, individual and community-level variables are included and assessed in an all-or-nothing fashion. I do not see the point of this. A logical way to build a 'best' model would be to have the variables with p < 0.2 as potential candidates, and then use a forward selection in the null model or backward selection in model iii to choose the best set of variables. Things like size of child at birth and age of child are nowhere significant - why should a so-called 'best' model include them? Statistically, a 'best' model should cut down as much slack as possible without removing true associations.

2.c. All the models are simple association models - that too with data from different years for different countries. There is no temporal factor, nor any possibility of inferring any kind of causality from such data. As such, statements like "maternal age affects infant feeding practices" (line 279) may be true, but are beyond the scope of this work. The only thing that can be said here is something in the lines of what the authors say immediately after this - "Those mothers older than 25 years were more likely to utilize formula feeding as compared to those younger." All statements in the discussion section should avoid using terms like affect/impact or likewise. Non-causal statements maintained by association and likelihood are the only possible inferences here.

3. A huge component of the analysis is constituted by the sample weights that the authors mention in many places but never provide any information on.

3.a. How are these weights constructed? Are they provided as part of the original survey dataset? If yes, what methods did they use to arrive at these weights?

3.b. The entire point of using survey weights is to adjust sample contributions in the perfect way. In this scenario, a sample may amount to more than a sample size of 1 if it is deemed to be substantially different from other data points. On the other hand, 10 very similar samples may end up being treated like a single sample if they are too similar or correlated. This begs the question: are the summaries in Tables 1-3 absolute summaries of the observations or are these weights taken into account there as well? If the former, then sadly those percentages do not mean anything in the grand scheme of things.

4. What is the purpose of Figures 1-2? They do not provide any new information beyond the summary tables.

Reviewer #3: Dear Author,

Thank you for the nice work. The discussion section is shallow and please discuss in detail with available published data elsewhere. Please update old reference. The rest of this work I found as very interesting

Reviewer #4: Thank you for doing this research in this area. It was well done and written with some minor corrections in a few sections of the paper. I have included the comments I have in the document attached, please.

Reviewer #5: This paper addresses an important topic, but I have some concerns about this study entitled “Determinants of formula feeding among mothers with infants and young children in six Sub-Saharan African countries: Multilevel analysis of data from demographic and health survey.”

In the abstract part of the manuscript line 31 “the prevalence of formula feeding among mothers with infants was 17.1%.” it is better to say The proportion of mothers with infants who use formula feeding was 17.1%."

In line 35 “higher odds was observed among plurality of child (AOR = 1.4; 95% CI (1.23-1.77))” What mean a plurality of children? Better to use the term Multiple Children.

In lines “26 &27” The data were taken from a recent demographic and health survey in six sub-Sahara African countries. Why only six countries? What is the reason behind that?

Given that more than Six countries have conducted similar surveys after 2019, the focus on only six countries might limit the generalizability of the findings.

The conclusion: the conclusion lacks clarity and specificity: specify why formula feeding is a concern (e.g., health risks, economic factors, cultural issues). Additionally, lacks consistency and uniformity: make sure the suggestions make sense in light of the identified issue.

Furthermore, it has to be Implementable Suggestions: Make clear the steps that policymakers and stakeholders should take.

in the discussion section line 265 -266, The prevalence of formula feeding might be lower in large-sample studies than in basic studies, do you have pieces of evidence? If you have evidence it must be cited

The discussion section especially for the second objective is shallow and lacks appropriate citations.

Consider starting with a summary of your main findings before diving into specific associations.

In lines 301-305 Maintain consistent terminology when referring to groups. For example, use "educated mothers" and "mothers with no formal education" consistently throughout the section.

Please Clearly state the observed association and provide potential reasons supported by the literature.

Why do you fail to explain why community literacy levels might impact individual feeding practices?

I also recommend you show the spatial distribution.

The manuscript contains several language issues that need to be addressed to improve clarity and readability, such as the Sentence Structure, Voice, Verb Tense, and Punctuation.

6. PLOS authors have the option to publish the peer review history of their article (what does this mean?). If published, this will include your full peer review and any attached files.

Reviewer #1: No

Reviewer #2: **Yes: **Rupam Bhattacharyya

Reviewer #3: No

Reviewer #4: No

Reviewer #5: No

---

## [Author Response · Author response to Decision Letter 0]

10 Aug 2024

Dear Editor and Reviewers, thank you very much for your valuable and thoughtful comments, suggestions, and recommendations. We have accepted and appreciated all your comments and suggestions to our manuscript. We have addressed all the comments by providing point by point response and correction to the manuscript. Your comments are crucial to produce high quality paper. We have significantly revised and improved the quality of the manuscript guided by your comments, suggestion, and recommendations. Here below are presented the responses of your comments for each point.

---

## [Decision Letter · Decision Letter 1]

16 Sep 2024

PONE-D-24-24055R1Determinants of formula feeding among mothers with infants and young children in six Sub Sahara African countries: Multilevel analysis of data from demographic and health surveyPLOS ONE

Dear Dr. Ali,

Thank you for submitting your manuscript to PLOS ONE. After careful consideration, we feel that it has merit but does not fully meet PLOS ONE’s publication criteria as it currently stands. Therefore, we invite you to submit a revised version of the manuscript that addresses the points raised during the review process.

We look forward to receiving your revised manuscript.

Kind regards,

Dev Ram Sunuwar, MS

Academic Editor

PLOS ONE

Journal Requirements:

Reviewers' comments:

Reviewer's Responses to Questions

**Comments to the Author**

1. If the authors have adequately addressed your comments raised in a previous round of review and you feel that this manuscript is now acceptable for publication, you may indicate that here to bypass the “Comments to the Author” section, enter your conflict of interest statement in the “Confidential to Editor” section, and submit your "Accept" recommendation.

Reviewer #1: (No Response)

Reviewer #2: All comments have been addressed

2. Is the manuscript technically sound, and do the data support the conclusions?

Reviewer #1: Partly

Reviewer #2: Yes

3. Has the statistical analysis been performed appropriately and rigorously? 

Reviewer #1: Yes

Reviewer #2: Yes

4. Have the authors made all data underlying the findings in their manuscript fully available?

Reviewer #1: Yes

Reviewer #2: Yes

5. Is the manuscript presented in an intelligible fashion and written in standard English?

Reviewer #1: Yes

Reviewer #2: Yes

6. Review Comments to the Author

Reviewer #1: Comments on statistical significance between institutional delivery, maternal educational level and media coverage has not been addressed properly. Do not just link it with your logic, add scientific reference contrary to your findings.

Reviewer #2: Thank you for the detailed updates to the manuscript. I have no issues to accept the manuscript in its current form.

7. PLOS authors have the option to publish the peer review history of their article (what does this mean?). If published, this will include your full peer review and any attached files.

Reviewer #1: No

Reviewer #2: No

---

## [Author Response · Author response to Decision Letter 1]

22 Sep 2024

Response to reviewer comments 

Reviewer #1: Comments on statistical significance between institutional delivery, maternal educational level and media coverage has not been addressed properly. Do not just link it with your logic; add scientific reference contrary to your findings.

Author’s response: Dear reviewer, thank you very much for your insightful feedback. You are correct that statistical significance does not always fully capture the complexity of real-world outcomes. We have made amendments on the discussion section of the manuscript. While factors such as maternal education, institutional delivery, and media exposure were statistically linked to bottle feeding in our study, we acknowledge that these associations may not directly translate into practical or policy-related justifications without considering broader contextual factors. For instance, the promotion of institutional delivery may aim to improve maternal and child health outcomes, but it does not necessarily mitigate the influence of cultural practices or socioeconomic conditions that drive formula feeding. This underscores the need for a nuanced interpretation of findings that integrates both statistical significance and the broader public health context, as emphasized in similar studies. Other similar studies also proved our finding cited in the discussion section of the manuscript. 

Furthermore, causal inference about the linkage between formula feeding and the above factors would require additional country-specific qualitative and quantitative studies. We have stated recommendation for future researchers on the discussion section of the manuscript. 

Reviewer #2: Thank you for the detailed updates to the manuscript. I have no issues to accept the manuscript in its current form.

Author’s response: Dear reviewer, thank you very much

---

## [Editor Report · Decision Letter 2]

30 Sep 2024

Determinants of formula feeding among mothers with infants and young children in six Sub Sahara African countries: Multilevel analysis of data from demographic and health survey

PONE-D-24-24055R2

Dear Dr. Ali,

We’re pleased to inform you that your manuscript has been judged scientifically suitable for publication and will be formally accepted for publication once it meets all outstanding technical requirements.

Kind regards,

Dev Ram Sunuwar, MS

Academic Editor

PLOS ONE
---

## [Editor Report · Acceptance letter]

8 Oct 2024

PONE-D-24-24055R2 

PLOS ONE

Dear Dr. Ali, 

I'm pleased to inform you that your manuscript has been deemed suitable for publication in PLOS ONE. Congratulations! Your manuscript is now being handed over to our production team.

Kind regards, 

on behalf of

Mr Dev Ram Sunuwar 

Academic Editor

PLOS ONE